# Monocyte Differentiation and Heterogeneity: Inter-Subset and Interindividual Differences

**DOI:** 10.3390/ijms24108757

**Published:** 2023-05-15

**Authors:** Helen Williams, Corinne Mack, Rana Baraz, Rekha Marimuthu, Sravanthi Naralashetty, Stephen Li, Heather Medbury

**Affiliations:** 1Vascular Biology Research Centre, Department of Surgery, Westmead Hospital, Westmead, NSW 2145, Australia; cmac3879@uni.sydney.edu.au (C.M.); rana.baraz@health.nsw.gov.au (R.B.); rmar9970@uni.sydney.edu.au (R.M.); snar8484@uni.sydney.edu.au (S.N.);; 2Sydney Medical School, The University of Sydney, Westmead, NSW 2145, Australia; 3Chemical Pathology, NSW Health Pathology, Westmead Hospital and Institute of Clinical Pathology and Medical Research, Westmead, NSW 2145, Australia; 4. Blacktown/Mt Druitt Clinical School, Blacktown Hospital, Western Sydney University, Blacktown, NSW 2148, Australia

**Keywords:** monocyte, differentiation, heterogeneity, monocyte subsets, inflammation, lipid, trained immunity

## Abstract

The three subsets of human monocytes, classical, intermediate, and nonclassical, show phenotypic heterogeneity, particularly in their expression of CD14 and CD16. This has enabled researchers to delve into the functions of each subset in the steady state as well as in disease. Studies have revealed that monocyte heterogeneity is multi-dimensional. In addition, that their phenotype and function differ between subsets is well established. However, it is becoming evident that heterogeneity also exists within each subset, between health and disease (current or past) states, and even between individuals. This realisation casts long shadows, impacting how we identify and classify the subsets, the functions we assign to them, and how they are examined for alterations in disease. Perhaps the most fascinating is evidence that, even in relative health, interindividual differences in monocyte subsets exist. It is proposed that the individual’s microenvironment could cause long-lasting or irreversible changes to monocyte precursors that echo to monocytes and through to their derived macrophages. Here, we will discuss the types of heterogeneity recognised in monocytes, the implications of these for monocyte research, and most importantly, the relevance of this heterogeneity for health and disease.

## 1. Introduction

Studies during the coronavirus disease 2019 (COVID-19) pandemic suggested the wide spectrum of disease severity from asymptomatic to hyperinflammation, multisystem failure, and death is regulated by innate cellular immunity and in particular, correlated with the morphology and immunophenotype of monocytes [1,2,3]. There is also evidence showing that epigenetic programming of monocytes has implications in disease processes such as atherosclerosis, inflammation, autoimmunity, and sepsis, and in vital functions such as immunity and healing [4,5]. Such a wide contribution to disease brings monocytes to the fore. In humans, monocytes comprise about 2–8% of white blood cells [6]. They are produced from haematopoietic stem and progenitor cells (HSPC) in the bone marrow, circulate in the blood for about 1–3 days, and then migrate into tissues and differentiate into macrophages or dendritic cells [7]. Indeed, the influx of monocytes in response to inflammation is an important contributor to macrophage presence in tissue [8]. There are three main subsets of monocytes now identified in humans: classical (CD14++CD16−), intermediate (CD14++CD16+), and nonclassical (CD14+CD16++). Researchers have discovered that these subsets show some functional differences, such as their inflammatory responses and migratory potential [9,10,11]. Monocyte functions, either as a whole or of specific subsets, are altered in several disease states [12,13]. This has hinted that the functions of the subsets may not be tightly preserved but blurred in disease states. However, interestingly, this blurring may not just be restricted to disease. Changing the frame of reference to the individual, it has become apparent that even in generally healthy individuals, subset differences are overridden by interindividual variation [14]. This functional plasticity of the cells likely arises in the bone marrow and is recapitulated as the cell differentiates through to a macrophage. In this review, we will discuss how monocyte subsets arise, how they are identified, and their functions, including how these change in disease as well as the extent to which monocytes differ between individuals. We will also discuss whether the changes to monocytes occur prior to their release into the circulation and whether changes are recapitulated from precursors, through monocytes to the macrophages they form.

## 2. Monocyte Ontogeny

### 2.1. Monocyte Development

Like most blood cells, monocyte ontogeny originates from HSPC predominantly in the bone marrow. Haematopoietic stem cells (HSC) are pluripotent cells with a self-renewing capacity that is lost upon differentiation into multipotent progenitors (MPP), which in turn reduce their multipotency as they differentiate through a hierarchy of increasingly lineage-committed progenitors [15]. The traditional model describes MPP progeny as being divided into common myeloid progenitors (CMP) or common lymphoid progenitors (CLP) [16,17], monocytes being of the myeloid lineage. CMP then differentiate further into granulocyte-macrophage progenitors (GMP) which give rise to monocytes and granulocytes, or into megakaryocyte-erythroid progenitors (MEP) giving rise to erythrocytes, platelets, and megakaryocytes [16,17] (Figure 1).

However, more recent studies have identified additional intermediate progenitor populations. For example, a mouse study identified that separate GMP and monocyte-dendritic cell progenitors (MDP) independently give rise to a subset of neutrophil-like monocytes and a subset of monocytes capable of differentiating into dendritic cells, respectively [18,19]. The MDP differentiate into monocytes through an intermediate termed the common monocyte progenitor (cMoP) [20], which requires interferon regulatory factor-8 (IRF8) signalling for monocyte differentiation [21]. This hierarchy was confirmed by the isolation of GMP and an intermediate monocyte progenitor (MP) giving rise to monocytes, as well as MDP and their intermediate cMoP [22] (Figure 1). Meanwhile, other studies have completely challenged the traditional sequential differentiation of progenitors. For example, it has been proposed that the MPP step can be omitted and that HSC can directly produce a myeloid-restricted repopulating progenitor, of which some individual cells are already committed to a megakaryocyte lineage, while others maintain a common myeloid lineage potential [23]. Alternatively, a pre-granulocyte-macrophage progenitor has been identified that through one pathway gave rise to mast cells, eosinophils, megakaryocytes, and erythrocytes, and through another pathway gave rise to monocytes, neutrophils, and surprisingly, lymphocytes [24].

The more specific characterisation of these progenitor populations at the single-cell level (with the advent of techniques such as single-cell RNA sequencing (scRNA-seq)) has revealed greater heterogeneity within these haematopoietic progenitor populations that may further challenge the traditional hierarchical model of differentiation. A study of myeloid progenitor cells in mice revealed that they could be divided into multiple distinct subpopulations based on gene expression profiles, some of which matched marker genes for monocyte, neutrophil, or erythrocyte progenitors [25]. Another study could distinguish separate sub-populations of CMP based on expression levels of the PU.1 transcription factor, which independently produced precursors for GMP or MEP [26]. This suggests that even earlier myeloid progenitor populations are heterogeneous with distinct clusters of cells perhaps already committed to specific cell types. Furthermore, scRNA-seq has also revealed that all of the cells derived from an individual MPP cell will usually be of the same differentiated cell type [27], further supporting multiple distinct cell profiles within the MPP population. However, other scRNA-seq studies have confirmed that some early progenitor populations do still have a gene profile indicative of multi-lineage potential [28]. Therefore, differentiation of blood cells from HSPC, including monocyte differentiation, may be better considered a continuum of cell phenotypes and gene profiles with multiple branching points, rather than separate hierarchical stages of distinct progenitor populations [29].

### 2.2. Monocyte Mobilisation into the Circulation

The differentiation of monocytes from HSC through the progenitor populations described above predominantly occurs in the adult bone marrow and is promoted by the macrophage colony-stimulating factor (M-CSF) [30]. However, monocytes must then be mobilised to the blood where they circulate or migrate into tissue, often to sites of infection or injury. In the context of infection, this mobilisation has been demonstrated in mice in response to circulating Toll-like receptor (TLR) ligands inducing cells in the bone marrow to express monocyte chemoattractant protein-1 (MCP-1), which binds to C-C chemokine receptor type 2 (CCR2) on the monocyte cell surface and triggers their release into the circulation [31]. Of the three monocyte subsets, CCR2 is most highly expressed on the classical subset [14,32], and as such, it is largely assumed that they are the first to be released from the bone marrow. Indeed, as only classical monocytes are observed in a human bone marrow biopsy [33], the classical monocytes must be the main subset released from bone marrow with the intermediate and nonclassical monocytes presumably arising once the cells have entered the circulation. Further expansion of the monocyte population in the circulation is supported by the identification and isolation of a subpopulation of human monocytes that proliferate when cultured with M-CSF or granulocyte macrophage colony-stimulating factor (GM-CSF) [34,35,36].

Human studies using the administration of deuterium-labelled glucose have traced monocyte mobilisation from the bone marrow. The labelled glucose was first detected only in classical monocytes (around 3–4 days after administration) indicating classical monocytes are the first to proliferate, as the labelled glucose is incorporated into rapidly dividing cells [37]. The label was only detected in intermediate monocytes on day four and nonclassical monocytes on day eight, suggesting that these subsets are not simultaneously released from the bone marrow with the classical subset, but that they emerge from the classical monocytes after their initial production and release. The data fit a model demonstrating that classical monocytes are released from the bone marrow and then differentiate into intermediate monocytes. The intermediate monocytes finally differentiate into nonclassical monocytes after a delay, with this differentiation perhaps occurring not only in the blood but also once the intermediate monocytes have left the circulation.

Another deuterium-labelled glucose tracing experiment showed that in a steady, homeostatic state, classical monocytes reside in the bone marrow for 1.6 days before mobilising to the bloodstream, where they circulate for 24 h [33]. Most leave the circulation and extravasate into the tissue or die, with approximately 1% of the cells differentiating into intermediate monocytes in the circulation [33]. These intermediate monocytes circulate for 4.3 days before the entire population differentiates into nonclassical monocytes. The nonclassical monocytes circulate for approximately 7.4 days before being recruited to patrol the vascular endothelium, leaving the circulation, or dying (Figure 1). The direct differentiation of the classical monocytes into the intermediate and then nonclassical monocytes was confirmed by grafting human classical monocytes into a mouse model with a humanised system of mononuclear phagocyte development [33]. After 24 h, the grafted cells had assumed an intermediate phenotype (as determined by whole blood flow cytometry), followed by a nonclassical phenotype after 96 h. This model of sequential differentiation through the three monocyte subsets has been further supported in models of human experimental endotoxemia which induces complete depletion of monocytes in the circulation [33,38]. As monocytes were restored in the circulation (after deuterium labelling), again it was the classical phenotype that was evident first after 4 h, followed by intermediate monocytes reported at 6–8 h [38] or 24 h [33] and nonclassical monocytes after 24 h [33,38]. Therefore, it was observed that classical monocytes are mobilised faster from the bone marrow and differentiation through the subsets occurs over a shorter timespan in a state of inflammation, compared to in homeostasis.

It is also important to note that monocytes not only migrate from the bone marrow to the circulation and then into the tissue, but in mice it has been shown that migration also occurs in the reverse direction. Transendothelial migration of monocytes has been observed from the vascular wall directly back into the circulation [39,40,41] or via drainage to lymph nodes [42]. Furthermore, murine monocytes can even migrate from the circulation back into the bone marrow to preserve their lifespan during periods of prolonged fasting [43].

While the majority of monocyte production and mobilisation to the circulation occurs from the bone marrow, murine studies suggest that an additional reservoir of monocytes is located in the spleen in adults [44,45]. These monocytes are able to be rapidly recruited to sites of infection, injury, or inflammation to enhance or supplement the bone marrow response. However, unlike the bone marrow, both classical and nonclassical monocytes are found in the spleen so both can be rapidly deployed to the circulation from this site [45]. While monocytes are also present in the human spleen [46], it has not yet been specifically investigated whether they also form a supplementary reservoir deployed for infection and injury response, as seen in mice.

### 2.3. Emergency Haematopoiesis as an Adaptive Mechanism of HSCs against Inflammatory Agents

HSPCs, like their immune cell lineages, express Toll-like receptors (TLRs), and cytokine and growth-factor receptors (receptors for interleukin (IL)-1β, IL-6, M-CSF, and type I and II interferons) [47] that enable them to directly respond to external or internal pathogen-/damage-associated molecular pattern (PAMP and DAMP) stimuli associated with infections or chronic inflammatory conditions [47]. Interestingly, different stimuli may lead to different myelopoietic responses and the mechanistic effects of distinct receptor binding and signalling on the regulation of HSC and progenitor cells have been reviewed extensively by Chavakis et al. [47]. Importantly, these stimuli can act directly on the HSC or the MPP, and the CMP [48]. HSCs are normally dormant, but their exposure to inflammatory mediators induces transcriptomic changes that orchestrate an enhanced proliferation of the cells in a process called ‘emergency myelopoiesis’, which induces preferential myeloid over lymphoid development [49]. Additionally, rapid recruitment of the monocytes to sites of injury also stimulates direct and indirect molecular signals resulting in a state of ‘emergency’, leading to the expansion of the CMP and GMP from the HSPC to replenish the monocyte pool in the circulation [50]. This shows that HSC are both ‘pulled’ by peripheral monocyte deficiencies and ‘pushed’ towards cell division and differentiation induced by inflammation. Expansion of HSPC and monocytosis is a typical response in dyslipidaemia and cardiovascular disease (CVD) [13]. Mechanistically, low-density lipoprotein cholesterol (LDL-C) increases CD34+ HSPC levels in the circulation by directly inducing their proliferation and through the cytokine IL-17 and G-CSF axis [51].

## 3. Classification of Monocyte Subsets

### 3.1. Monocyte Subset History

Currently, in humans, that there are three main monocyte subsets is widely accepted. However, this has not always been the case and indeed, additional subsets are being identified. Monocytes were first discovered in 1926 (Table 1) followed by their identification as an immature member of the reticuloendothelial system (RES), making up 3–7% of total leukocytes [52]. Prior to the 1980s, monocytes were only able to be broken into subsets based on centrifugation [53] by their size and density [53,54]. Even with this traditional approach, prior to the discovery and use of flow cytometry, multiple subtypes -2, 3, and even 4- of monocytes were proposed. Meuret et al. (1974), divided blood monocytes into three groups according to their nuclear morphology, with type a (round oval) being the most immature, followed by type b (slightly folded) and type c (distinctly folded) being the most mature form of monocytes in the blood [54] (Table 1). Thereafter, Barret et al. (1979) used cytochemical stains and Complement 3 receptor assays to suggest that monocytes consist of at least 1–4 subpopulations due to their Fc and complement C3 cell surface receptor reactivity patterns [55]. Further work was done and monocytes were separated into Fc receptor positive, (FcR+), and FcR- subsets using rosetting, density gradient centrifugation, and adherence [56]. In addition, CD64 (Fcγ receptor I) negative monocytes were identified as a minor human monocyte subset that displayed higher accessory cell capacity in antigen-driven T cell activation and antiviral activities than CD64+ monocytes [57]. Functional investigations of these subsets gave way to surface marker studies in the 1980s, with Passlick et al. taking advantage of two-colour immunofluorescence to report that, besides the strong positive CD14++ population, there was also a CD14+CD16+ subset [58] (Table 1).

Early in the 1990s, the classification of monocytes to these two subsets, defined by CD14 and CD16, was used by many groups examining monocyte population changes in disease states such as sepsis [59], human immunodeficiency virus (HIV) infection, [60], and in patients with acute and chronic infections undergoing hemodialysis [61] (Table 2). This shows how two subsets was the consensus at this time. These studies shed light on the pivotal role of the CD14+CD16+ subset in disease progression. Leaping into the 2000s, more work was done on delineating the monocyte subsets further into three distinct subpopulations. These subsets were identified as CD14++CD16−, CD14++CD16+, and CD14+CD16++ (Table 2), with the importance of the CD14++CD16+ in disease states such as asthma being recognised [62]. A tremendous bulk of work had been done and yet the subsets were not given a proper name to go by.

In 2010 the Nomenclature Committee of the International Union of Immunologic Societies ‘officially’ recognised and named the three subsets: Classical (CD14++CD16−), Intermediate (CD14++CD16+), and Nonclassical (CD14+CD16++) [63]. This unified the naming strategy given to the monocyte subsets across the immunological scientific platform, with these names being adopted in clinical research (Table 2). Having flow cytometry as the gold standard for identifying the subsets, more advanced techniques have emerged to assess their phenotype, including microarray [64], and SuperSAGE transcriptome analysis [65] to assess gene expression and Cytometry by time of flight (CyTOF) mass cytometry to assess multiple surface markers on individual cells [66]. Transcriptomic studies provided a deeper insight into the similarity between the intermediate and nonclassical subsets. Thomas et al. [66] showed the efficacy of the CyTOF mass cytometry approach (using 25 colours to measure protein abundance at a single-cell resolution) and the tSNE algorithm, to display the cells in a 2-dimensional scatter plot using information from each marker used.

**Table 1 ijms-24-08757-t001:** Categorisation of monocyte subsets in health.

No of Subsets	Name of Subsets	Identifying Markers	Method Used	Reference	Year
1	Monocytes Monocytes	No markers used No markers used	Immunohistochemistry staining, phagocytosis Cytochemistry, light microscopy, Biochemical assays	[67] [52]	1926 1966
2	Monocyte large fraction (80–90%) Intermediate small fraction (12–18%) Monocytes	Esterase & Fc receptor detection CD14++ CD14+/CD16+	Elutriation. Cell fractions determined by peroxidase and esterase staining, neutral red phagocytosis, and iron particle phagocytosis 2 colour fluorescence flow cytometry & cell sorting	[53] [58]	1979 1989
	Round oval (type a) Slightly folded (type b) Distinctly folded (type c)	Measured Chloroacetate-Esterase, Acetate-Esterase and 3H-TDR labelling index	Cytochemical reactions and DNA-Synthesis activity	[54]	1974
	Monocytes	Detection of Fc & Complement receptors	Cytochemical stains, C3 receptor assays	[55]	1979
3	Classical Intermediate Nonclassical	CD14++CD16− CD14++CD16+ CD14+CD16++	Flow cytometry	[63]	2010
	Classical Intermediate Nonclassical	CD14++CD16− CD14++CD16+ CD14+CD16++	Microarray, flow cytometry and cytokine production	[64]	2011
	Classical Intermediate Nonclassical	CD14++CD16− CD14++CD16+ CD14+CD16++	SuperSAGE transcriptome analysis	[65]	2011
	Classical Intermediate Nonclassical	CD14++CD16− CD14++CD16+ CD14+CD16++	CyTOF mass cytometry	[66]	2017
4	Mono1 (classical) Mono3 Intermediate) Mono4 (intermediate) Mono2 (nonclassical)	CD14++CD16− CD14++CD16+ * CD14+CD16++	Single-cell RNA sequencing (scRNA-seq)	[68]	2017

* Within this subset, single-cell RNA-seq identified heterogeneity, namely two distinct clusters which were assigned Mono3 and Mono4. Gene expression profiling showed Mono3 expressed cell cycle and trafficking genes, while Mono4 expressed a cytotoxic gene signature.

**Table 2 ijms-24-08757-t002:** Utilisation of established monocyte subsets in disease states.

No of Subsets	Disease	Identifying Markers	Method Used	Reference	Year
	Sepsis	CD14++ (regular) CD14+CD16+	2 and 3-colour immunofluorescence, cell sorting	[59]	1993
2	HIV/AIDS	CD14highCD16low CD14lowCD16high	Flow cytometry	[60]	1995
	Haemodialysis	CD14++ CD14+CD16+	Flow cytometry, phagocytic activity	[61]	1998
	Haemodialysis	CD14++CD16− CD14++CD16+ CD14+CD16+	Flow cytometry	[62]	2008
3	Asthma	CD14++CD16− CD14++CD16+ CD14+CD16+	Flow cytometry	[69]	2008
	Chronic Kidney disease	CD14++CD16− CD14++CD16+ CD14+CD16+	Multi-colour flow cytometry	[70]	2018

Their revised gating scheme increased the purity of both the intermediate and nonclassical monocyte subsets from 86% to 98.8%, and 87.2% to 99.1%, respectively. Applying this revised gating strategy to high and low coronary artery disease (CAD) individuals improved monocyte subset classification in CVD. While new methods confirm these three subsets and show that including additional markers may better define the subsets, practically speaking, this refined approach would be difficult to implement across clinical studies. Furthermore, a scRNA-seq study by Villani et al. (2017) [68] shed more light on the heterogeneity of the intermediate monocytes, thus distributing the monocytes into four subsets: classical as Mono1, intermediates as Mono3 and Mono4, and nonclassical as Mono2. Whether monocytes continue to be classified into the three currently recognised subsets, or whether the existence of this continuum will result in new ways of classifying these cells remains to be seen.

### 3.2. Monocyte Gating in Flow Cytometry

With the standardisation of the three monocyte subsets’ nomenclature, flow cytometry-based classification is now extensively used in clinical studies. However, inconsistencies in monocyte gating strategy exist between studies. This complicates efforts to enumerate the subset proportions and understand their functions. There is no consensus on monocyte gating, making it difficult to assign specific functions to each subset. This lack of consensus may be due to monocyte subsets not existing as distinct populations but rather as a spectrum [64,71] (discussed in Section 3.3). Using flow cytometry, numerous approaches have been employed to get rid of contaminating cells and distinguish the monocytes into three subsets. Monocyte gating steps begin with different ways to eliminate clumps and debris. Next, monocytes are selected either by drawing a tight gate around monocytes in a forward scatter vs. side scatter plot [72,73] or by using an additional monocyte-specific marker (other than CD14 and CD16) e.g., HLA-DR or CD86. This excludes contaminating cells, particularly neutrophils, T cells, B cells, and natural killer cells [74,75]. On top of the gating approaches used to obtain pure monocytes, strategies to demarcate the subsets differ in the type and placement of gates (differing in the starting and ending of each subset) leading to variability in the enumeration of subset proportions across studies.

While the classical monocytes, as the major population, may be reasonably straightforward to gate, the separation of the intermediate and nonclassical populations is highly variable. Different shapes of gates such as quadrant, rectangular, or trapezoid, have been used in studies to distinguish between the subsets. So different are the gating methods that studies have directly compared methods to understand their impact on findings. A study comparing rectangular and trapezoid gating (Figure 2a,b) was conducted on over 400 people with chronic kidney disease. Both strategies were sensitive enough to detect high proportions of intermediate monocytes in people who went on to have a clinical cardiovascular event [76]. While both techniques were sufficiently sensitive, rectangular gating is more widespread and allows a better comparison between different individual studies (Figure 2a,b) [76,77,78]. Even where the shape of the gates drawn is consistent, the exact placement of gates also differs between studies, as in the case of division between classicals and intermediates which varied widely among discrete studies. To make the distinction between these subsets more objective, a few methods have been utilized. Some studies recommend the use of an isotype control for CD16 to set clear demarcation for the ending of classical subsets and starting of intermediates [78]. In our studies, we employed a combination of the rectangular gating approach with data displayed in a zebra plot which provided additional visualisation cues to more precisely and objectively distinguish each subset (Figure 2c) [74]. Some studies recommend the use of additional markers such as CCR2 which is highly expressed by classicals [32] and SLAN (expressed by nonclassicals) [79] for correct enumeration of subsets. Using SLAN is a widely accepted and practical way of distinguishing non-classicals from intermediates [79]. More complex methods to accurately gate the three subsets have also been examined. A recent study analysed a combination of 33 markers using CyTOF to delineate monocyte subsets instead of the conventional CD14 vs. CD16 combination and found a panel of five markers (CD33, CD64, CD86, HLA-DR, and CCR2) that could distinguish the three subsets, eliminating all contaminating cells even after in vitro stimulation [80]. Various automated computational approaches, such as tSNE/viSNE and SPADE, have also been used to cluster monocyte subsets and have increased the accuracy of the gating methods employed [66,81]. As such, while many gating methods are sufficiently sensitive to detect monocyte alterations in disease states, comparisons between studies in delineating monocyte proportions and functions remain difficult due to a lack of consistency.

### 3.3. Monocyte Spectrum

The differences in gating strategy between studies seem to arise from monocyte subsets being neither distinct nor homogeneous. While characterising the phenotype of monocyte subsets with flow cytometry, Hijdra et al. subdivided the monocyte plot into 10 gates based on an increasing CD16 expression and a decreasing CD14 expression. The change in the expression pattern of markers such as tumor necrosis factor receptor (TNFR)1, HLA-DR, and CCR2 was gradual. For CCR2, its highest expression was in the classical subset but then it slowly faded out as the gates progressed through the intermediates and the nonclassical subsets [71]. This introduced the idea of a spectrum or a continuum of several monocyte markers across the different subsets, with a gradual increase of some or a gradual decrease of others. Hamers et al. delved into this heterogeneity using mass cytometry and clustering algorithms. They identified eight monocyte subsets: four within the classical subset, three within the nonclassical subset, and lastly, the intermediate population. While some markers, such as SLAN, were expressed only on nonclassical monocytes, many markers were expressed by several subpopulations, although this was to different degrees [82]. In the same year, while investigating how best to identify the subsets, Ong et al. showed several markers that were expressed in a gradually increasing or gradually decreasing manner across the subsets, solidifying the continuum phenomenon [80]. In alignment with this, we assessed the expression of CD11b, CD11c, and CCR2 across the monocyte subset continuum. Using a heat map, we found that CD11b (Figure 3) and CD11c displayed differential expression within monocyte subsets, which was particularly evident within the classical subset, while CCR2 expression followed in the traditional classical, to intermediates, to nonclassical continuum [14] which suggests that cells may take different trajectories as they mature.

This shift of monocytes “along their continuum” has also been examined in a cohort of 227 patients with high cardiovascular risk [83]. The continuous distribution of CD14 and CD16 fluorescence was evaluated within each subset. Differing from the traditional gating method, the authors ventured further and divided the classical subset into CD14++CD16− and CD14++CD16dim cells which displayed clear differences among CAD patients. In these patients at the baseline, there was a doubling of CD14++CD16+ intermediate monocytes and a shift of nonclassical and classical monocytes towards intermediates cells, suggesting a gradual shift in increasing CD16 both across subsets and with the disease. In addition, in patients with type 2 diabetes, there was a strong and consistent upregulation of CD16 *within each monocyte subset* versus nondiabetic patients, albeit with no change in the frequency or number of each subset [83]. To further support the idea that monocytes exist not in discrete subsets but as a continuum, a study by Cignarella et al. (2018) proposed three theoretical ways of analysing the monocyte continuum within the traditional CD14/CD16 plot, ultimately paving the way for fine-tuning the assessment of CVD risk in large patient cohorts [84]. The first of these three theoretical ways was explored in the study explained above [83], where the percentage of cells and MFI of CD14 and CD16 within each gate was reported to enable the exact transitioning of monocytes from one subset to another. The second way used multiple, discrete gates along the CD14/CD16 plot, further allowing a more detailed analysis of monocytes as they transition [71,84]. The third way was termed as the clock rule which displays the CD14/CD16 plot as a continuum along a 90° curve, capturing the expression of both markers as a single number. This latter approach is only theoretical and has not been validated yet.

## 4. Monocyte Subset Functions

### 4.1. In Vitro Evidence

As described earlier, monocytes play important roles in maintaining tissue homeostasis by initiating, propagating, and resolving immune responses against infection and injury [31]. Importantly, monocyte subsets play different roles which vary in steady state and under pathological conditions [85]. Classical monocytes play major roles in initiating innate immune responses, phagocytosis, and migration by expressing chemokines, scavenger receptors, and pro-inflammatory cytokines [85,86]. Intermediate subsets are associated with antigen processing and presentation, monocyte activation, inflammation, and differentiation [65]. Nonclassicals are unique from other subsets in their patrolling behaviour, specifically the surveillance of vasculature [9], and have been predicted to display FcR and complement-mediated phagocytosis [64,85].

#### 4.1.1. Monocyte Subsets Differ in Their Expression of Functional Markers

Monocyte subsets express different surface markers at varying levels reflecting their functions (Table 3). Classical monocytes show high expression of phagocytic and scavenger receptors CD64, CD163, and CD36 compared to the other two subsets. This is linked to functions such as the initiation of inflammatory responses, phagocytosis, and the resolution of inflammation. Though classical monocytes are generally considered less inflammatory [87], they were found to express both inflammatory (CD64) and anti-inflammatory (CD163) markers in high levels, demonstrating their potential role in innate immune response, tissue remodelling, and the resolution of inflammation [4,9,64,85]. Intermediates express high levels of HLA-DR, CD80, CD86, and TNFR1, indicating their role in antigen presentation and inflammatory functions [64,88]. On the other hand, nonclassical monocytes show high expression of SLAN, CD115, siglec10, and TNFR2 which demonstrates their role in inflammation [85,88]. Monocyte subsets also differ in the expression of chemokine receptors and adhesion markers, implying that the subsets are likely to differ in their migration capacity and response to chemokines. Classical monocytes highly express CCR2 which potentially enhances their adhesion and migratory capacity to inflamed tissues whereas intermediate monocytes display higher expression of the chemokine receptor CCR5 than classical monocytes [64,89]. Additionally, classicals and intermediates express CD62L and CD11b in high levels, markers that aid in adhesion and migration in response to inflammation [9,90]. Nonclassical monocytes express high levels of CX3C motif chemokine receptor (CX_3_CR)1 and CD11c which may indicate their high potential to migrate to the vessel wall in response to inflammation [91]. Intermediates also express high CX_3_CR1 which primes them for trans-endothelial migration.

#### 4.1.2. Monocyte Subset Functional Heterogeniety Is Seen at Transcriptomic Level

Knowledge of subset functional heterogeneity is extended by gene expression profiling. While phenotypic characteristics are mainly derived from flow cytometric surface expression analysis, transcriptomic studies provided more profound characterisation in a quantitative and qualitative manner. After the consensus of the nomenclature publication, three major studies dived in to explore the genetic distinctions between the three monocyte subsets, with predicted biological functions validated by functional assays [9,64,65]. The most notable difference was the association of intermediates with antigen processing and presentation functions, as they highly expressed a cluster of major histocompatibility complex (MHC) Class II-restricted genes. This aligns with the high HLA-DR surface expression, along with the expression of the CD40 co-stimulatory molecule which aids in T cell stimulation [64,79]. Nonclassicals, like intermediates, have also been implicated in antigen processing and presentation as they express MHC-restricted genes, indicating that while intermediates and nonclassicals play critical roles in the adaptive immune system, classical monocytes do not [64,65,92]. That intermediates and nonclassicals differed in their expression of a class of MHC-restricted genes indicates a functional difference. Intermediates were found to express high levels of class II MHC genes, e.g., *CD74, HLA-DR* [65,79] which present a wide range of antigens to CD4 cytotoxic T cells [92,93]. Nonclassicals express MHC1-related genes which present peptides from viruses to CD8 T cells, aligning with the response of nonclassicals to viral nucleic acids [9,65].

While both intermediates and nonclassicals are associated with monocyte activation, they differ in their gene expression. Intermediates express genes that regulate chemotaxis and angiogenesis (*AIF1* and *TIE2*) as well as phagocytosis and tissue repair (*TGFB* and *CD93*) [94]. This indicates that despite their proposed inflammatory nature, intermediates are also likely involved in anti-inflammatory functions. Nonclassical monocytes show high expression of *CD16*, *HMOX1*, and *KLF1*, genes which reflect their dual inflammatory and anti-inflammatory roles in monocyte activation and differentiation [95,96]. On the other hand, the classical subset expresses gene clusters falling under angiogenesis, tissue repair, responses to a variety of cues (bacterial components, hypoxia, and toxins), and expression of pro-inflammatory mediators such as S100 proteins (A12, A8, A9), showing their ability to initiate immune responses against external cues and mediating tissue repair [64]. This aligns with their high expression of phagocytic and scavenger surface receptors as well the high CD163 marker expression.

#### 4.1.3. Cytokine Production Capability Differs between Monocyte Subsets

The in vitro functional assays to validate the transcriptomics data were then conducted using monocytes isolated from healthy individuals. Unfortunately, these functional studies yielded conflicting results. In terms of cytokine production, it was evident that all three subsets were capable of secreting cytokines in response to various stimuli, but differed in the response depending on the stimuli, dosage, and kinetics. In response to LPS stimulation, classical monocytes showed a mixed response, producing high levels of cytokines IL-6, IL-8, CCL2, and CCL3, whereas intermediates were the main producers of inflammatory cytokines (TNFα and IL-1β) in addition to producing IL-6 and CCL3 [9]. This evidence is in line with the previous reports on cytokine production by classicals and intermediates [97,98] showing their capability to respond to bacterial cues [9]. Nonclassicals did not respond to bacterial cues, however, they selectively produced pro-inflammatory cytokines such as TNFα, IL-1β, and CCL3 in response to viral nucleic acids and immune complexes, which may indicate a role in the pathogenesis of autoimmune diseases as well as against viral infection [9]. Interestingly, another functional study showed contradictory results with an enhanced secretion of inflammatory cytokines TNFα and IL-1β by nonclassicals in response to LPS and low or intermediary levels by intermediates [64]. This may be due to a difference in the dosage and time course of the experiments. However, classicals produced high levels of GM-CSF, IL-10, and IL-6, a mixed response consistent with the previous reports [64]. In line with the above finding, nonclassicals were found to be the most pro-inflammatory phenotype with high TNFα production basally and acute response to LPS stimulation [99]. As stated, different stimuli elicit different responses. Boyette et al. showed in response to agonists for TLR 1–9, all three subsets secreted pro-inflammatory cytokines IL-1β, IL-6, and TNFα [90]. Most notably, the classical subset was the highest producer of all three cytokines with TNFα levels being similar between classicals and intermediates [90] which is consistent with the intermediates secreting the bulk of inflammatory cytokines in response to TLR2 agonists [9]. Nonclassicals were found to secrete the lowest levels of cytokines [90]. A particular study using endothelial cell–monocyte coculture showed high production of IL-6 by classical monocytes and high TNFα by intermediates and nonclassicals, which was significantly reduced by the presence of classicals [100], indicating that the effect of other cells in the culture also impacts cytokine production. All these findings indicate, cytokine production is not clearly distinct among the subsets, moreover, different stimuli, (basal state or after a challenge), time course (acute and long term), and culture conditions are likely to impact the subset functions and lead to different immune responses.

In terms of phagocytosis and reactive oxygen species (ROS) production, clear differences exist with classicals and intermediates able to phagocytose latex beads and produce ROS—a function not seen in the nonclassical subset [9]—that is consistent with the degree of CD14 expression by the subsets. Interestingly, intermediates showed the highest ROS production in the unstimulated basal state, followed by nonclassicals and lastly, classicals [65]. Since ROS production is a critical response in inflammatory processes, this confirms the role of intermediates in inflammation.

While in vitro evidence supports functional differences between subsets, that is not to say the entirety of each subset is homogeneous in terms of function. It has been shown that sub-clusters exist within the subsets, and these differ in activation and differentiation status [86]. In addition, as monocytes alter their functions to respond to the microenvironment, functional differences in vitro may not fully reflect functions in health and disease.

### 4.2. In Vivo Evidence

Whether functions assigned to monocyte subsets from in vitro studies reflect their functions in vivo is important to consider. Unfortunately, in vivo functions, particularly in humans, remain difficult to study. Research in humans includes using ex vivo studies to infer in vivo function, labelling and tracking monocytes to understand their migration, and examining monocyte-derived cells in tissues obtained by biopsy or autopsy. In mice, adoptive transfer and selective depletion of specific subsets also give clues regarding subset functions. It is important to note that while mouse and human monocyte subsets broadly align (Table 4), the existence of some differences between them means they can’t be considered exact counterparts [9,101,102].

#### 4.2.1. Monocyte Subset Inflammation in Human Studies

Ex vivo studies in human disease states have predominantly implicated the intermediate and nonclassical monocytes as inflammatory, as they are elevated in numerous infections and inflammatory conditions (extensively reviewed elsewhere [85]). That the expansion of intermediate monocytes predicts poor outcomes, such as cardiovascular events, lends weight to this proposed inflammatory function [103]. Ex vivo studies have also used markers to indicate functions, with minimal processing techniques recommended for this venture [77]. In healthy controls, intermediate monocytes have lower CD163 than classical monocytes, with nonclassicals lower still [4]. As CD163 is associated with anti-inflammatory or tissue-remodelling macrophages [104], this could indicate classical monocytes are anti-inflammatory in the steady state. In parallel, CD86 which is associated with inflammatory M1 macrophages [104] is highest on nonclassical monocytes, lower on intermediate, and lowest on classical monocytes, which fits with the inflammatory functions attributed to intermediate and nonclassical subsets [4,105].

#### 4.2.2. Monocyte Subset Inflammation in Mouse Studies

While mouse studies cannot be directly translated to humans, they do permit the use of techniques that can’t be used on humans. Adoptive transfer of different monocyte subsets in mice seems to confirm their divergent inflammatory roles. Adoptive transfer of human classical monocytes into severe combined immunodeficiency (SCID) mice with tuberculosis resulted in a higher number of TGFβ+ and IL-10+ (anti-inflammatory) cells in the lungs [106], a finding which aligns with lower inflammatory cytokine release in response to bacterial cues in vitro. Transfer of CD16+ (intermediate and nonclassical) monocytes was associated with more TNFα+ (inflammatory) lung cells [106]. Even in mice lacking tuberculosis, these different functions were seen, with the number of TBFβ+ cells higher with the transfer of classical monocytes and TNFα+ cells higher when intermediate and nonclassical (CD16+) monocytes were transferred [106]. Nonclassical monocytes could certainly contribute to the development of rheumatoid arthritis. When nonclassicals (Ly6C− cells, the equivalent of human nonclassicals, Table 4) were absent in mice, rheumatoid arthritis did not develop, while the depletion of classical monocytes (Ly6C+) did not alter the development of arthritis [107]. In fact, classical monocytes may perform functions preventing arthritis, as adoptive transfer of these monocytes resulted in a clear delay in the development of arthritis [107]. In other cases, nonclassical monocytes may be beneficial. In cancer, mouse studies have revealed that nonclassical monocytes broadly perform more anti-tumoural functions while classical perform more pro-tumoural functions [108]. This highlights the need to consider functions in the context of the specific disease state. Unfortunately, little is known about the inflammatory functions of intermediate monocytes in vivo, which is likely due to them being a minor population in mice.

#### 4.2.3. Monocyte Subset Migration in Mouse Studies

In vitro studies suggest monocytes differ in their capacity to migrate to tissues. Efforts have been made to investigate these divergent roles in vivo and assess how these may change with disease, as their presence in the tissue can either promote or attenuate disease progression. Monocytes as a whole have been shown to enter both inflamed and non-inflamed tissue. Adoptive transfer of labelled monocytes into atherosclerotic (*APOE*^−/−^) mice resulted in monocytes in the heart and lungs and later liver and spleen, as well as in the inflamed atherosclerotic lesions [109]. Of note, migration to the aorta does not seem to occur in the absence of atherosclerosis, as no labelled cells were detected in the aortas of control (C57BL/6) mice [109]. Human studies have also reported the migration of monocytes to both inflamed and non-inflamed tissue. When labelled monocytes were re-infused into people with rheumatoid arthritis, monocytes tracked to their inflamed joints but were also detected in the lungs, spleen, bone marrow, urinary bladder, and bowel [110]. This supports that in a steady state, monocytes do enter numerous tissues and that in inflammation they track to the inflamed tissue. However, these studies do not permit the comparison of specific subsets regarding their migratory functions.

Examining murine subset differences, in the steady state, nonclassical monocytes patrol blood vessels, with imaging studies showing a crawling behaviour unique to this subset [111]. It appears nonclassical monocytes enter numerous tissues in the steady state, while classical monocytes do not. In healthy mice, adoptively transferred nonclassical monocytes were found in the spleen, blood, lung, liver, and brain, while classical monocytes were only detected in the spleen [10]. How the different subsets respond to inflamed tissue depends on the specific cause of inflammation and the environment milieu. In thioglycolate injection, classical monocytes tracked to the inflamed peritoneum in greater numbers than nonclassical monocytes, with the latter continuing to enter non-inflamed tissues [10]. In cancer, different studies have shown classical monocytes are the main subset to extravasate [108]. In atherosclerosis, the classical monocytes also preferentially accumulate in the inflamed tissue, as a greater number of classical monocytes were found than the nonclassical monocytes [11]. As the influx of monocytes into atherosclerotic plaques is an important step, classical monocytes are therefore likely to promote atherosclerosis. Nonclassical monocytes with their limited plaque influx are considered atheroprotective, supported by studies showing that mice lacking nonclassical monocytes readily develop atherosclerosis (reviewed [112]). In some cases, the nonclassical monocytes seem to be the first responders to infection and inflammation. In response to listeriosis or wounding, the nonclassical monocytes extravasated rapidly. The classicals extravasated later in response to listeriosis [111]. This rapid response of nonclassicals was deemed to be possible as patrolling monocytes are poised to quickly respond to inflammation. Indeed, labelled nonclassical monocytes were detected in the joints of arthritic mice but did not appear in the joints of the non-arthritic mice, implicating that nonclassicals are able to migrate in response to arthritis [107]. In a wound model, both monocyte subsets were directly recruited with classical monocytes, becoming nonclassical after extravasating [113].

#### 4.2.4. Monocyte Subset Migration in Human Studies

In humans during the steady state, the labelling and tracking of monocytes suggest that intermediate monocytes do not extravasate into the tissue, whereas classical and nonclassical monocytes likely do [33]. As in mice, migration responses to infection seem to depend on the disease state. Postmortem samples from people who had a recent acute myocardial infarction (AMI) indicated that the inflammatory phase (days after AMI) had more classical (CD16−) monocytes in the tissue, whereas at the proliferative phase (1–2 weeks after AMI) the intermediate/nonclassical (CD16+) and classical (CD16−) numbers were similar [114]. This supports that classical (CD16−) monocytes more readily migrate into inflamed tissue, but indicates CD16+ subsets appear later, once the inflammation has set in. Whether this is solely due to nonclassical migration or due to classicals becoming nonclassicals in situ is unclear. In people with rheumatoid arthritis, the nonclassical monocytes seem to show enhanced migration, similar to what was indicated in mouse models. The proportion of intermediate/nonclassical (CD16+) cells in synovial fluid was greater than that in blood, with this enrichment indicating the migration of nonclassical monocytes into inflamed joints [115]. As such, while monocytes are capable of migrating in both a steady state and in disease states, there are differences in the timing, frequency, and manner in which each subset does so. Overall, while determining in vivo functions of monocytes remains technically and conceptually challenging, there certainly appear to be differences between the subsets in terms of their functions, and these functional differences are more apparent in states of infection and inflammation.

## 5. Monocyte Heterogeneity-Shifting the Frame of Reference to the Individual

### 5.1. Monocyte Function Is Altered in Disease States

Numerous clinical studies find that monocytes (as a whole) have an increased inflammatory profile in various disease states or infections (e.g., HIV, malaria- for an extensive review see Rinchai [116]), but fewer studies have intentionally looked at the profile of specific monocyte subsets in these conditions. It is apparent that some changes occur across more than one monocyte subset, for example, in Rheumatoid arthritis, where gene expression of CCL2, IL-5, PPARγ, VEGF, TF, and IL-8 was increased in both the CD16+ and CD16− monocytes subsets (only two subsets were assessed), compared to healthy controls [117]. In COVID-19, all three subsets have increased levels of CD64, with classical and intermediates also having an increase in CD86 [105]. In head and neck cancer, increased levels of CD11b were seen on all three monocyte subsets, conversely, CX_3_CR1 was decreased in patients’ classical and intermediate monocytes [118]. Even in models where additional subsets have been described, consistent changes across the subsets are evident with the majority of the 11 monocytes subsets identified in Sjögren’s Syndrome having increased expression of TNFSF10 (TRAIL) compared to the 10 monocyte clusters identified in control subjects [119].

Very few studies have shifted the frame of reference further—to the individual. In the COVID-19 study (above) [105] several individuals with COVID-19 (moderate or severe) expressed higher levels of HLA-DR on all three of their monocyte subsets than that expressed by any subset for other individuals (patients and controls). In patients with severe COVID-19, this was also the case for CCR2. While these highlighted examples do not definitely demonstrate a blurring of the subset distinctions, as only a few markers were examined, the global transcription profiles of classical and intermediate monocytes have been found to become less distinct in children infected with dengue [120]. Moreover, distinct clustering was lost in monocytes from patients with severe disease or a secondary dengue infection. [120].

### 5.2. Monocyte Heterogeneity between Generally Healthy Controls

A degree of heterogeneity also exists between generally healthy individuals. Elderly patients (>65 years), for example, display a higher level of CX_3_CR1 on both classical and nonclassical monocyte subsets than younger adults (21–40 years) [121]. Significant differences have also been observed between Africans and Caucasians, with Africans exhibiting a higher expression of HLA-DR and CCR2 on classical monocytes [122] and Caucasians exhibiting a higher expression of CX_3_CR1 on a subgroup of intermediates monocytes. Though the two populations were different ethnicities, the differences were thought to be attributed to past exposure to parasites rather than genetics.

Examining interindividual differences, considerable differences between generally healthy individuals are apparent. Assessing a monocyte inflammatory state, we previously found that all three monocyte subsets of several individuals had a higher level of inflammatory markers, or cytokine production, than any monocyte subset of other individuals [4]. These differences were also detected at the single-cell level as seen on a flow cytometry dot plot overlay of one individual’s cells on another [4]. Similarly, (in another study [14]) we saw that recruitment marker (adhesion marker and chemokine receptor) expression was greater in some individuals compared to others—even for CCR2, CD62L, and CD49d which have been proposed to identify distinct/novel monocyte subsets [123]. Though CD49d expression was significantly higher on nonclassical monocytes than on either the intermediate or classicals (*p* < 0.01), there were three individuals (out of *n* = 30) whose CD49d expression was higher on their classical monocytes than on the nonclassical subset of 13 other individuals (Figure 4).

With the expression of one recruitment marker significantly correlating with that of most others (for all three monocyte subsets) [14] the distinct differences in the migratory ability of the monocyte subset—such as nonclassicals primarily having a patrolling phenotype—would be blurred. All monocytes in some individuals have an increased capacity to extravasate and potentially do so with a wider repertoire of recruitment markers than currently recognised for each subset. As it was noted that there was a significant inverse relationship between most of the recruitment markers and ApoA1, the protein associated with high-density lipoprotein cholesterol (HDL-C) (as well as several associations found with HDL-C), then increased extravasation may be occurring in individuals with dyslipidaemia [14]; functional studies are needed to confirm this.

Clearly, these flow studies are biased given that the markers were chosen by the investigators. Unbiased genomic studies could provide more insight as they obtain information on an array of pathways, however, they will need to be conducted on a greater number of people (in one study) to shed light as to whether there are high interindividual differences for other monocyte functions. The large number of chemokines and adhesion molecules assessed (in the flow studies), suggests that at least the migratory pathway is one that, though differing between the subsets, is heavily influenced by the microenvironment of each individual. As cell subset phenotype and function should be unique [124], then recruitment markers may largely reflect ‘functional states’—arising from cell plasticity—rather than ‘distinct subset’ characteristics. While further investigation is needed, the flow studies hint that, while on the one hand, there is great heterogeneity between monocytes, there is an element of which, for some individuals, the function of the subsets becomes more homogenous. While the blurring of the subset differences can be viewed as an increased overlapping of subsets on a Venn diagram of marker expression, the level of expression of the markers (for each subset) is also increased.

### 5.3. Baseline Monocyte Inflammatory State Is Determined in the Bone Marrow and Is Recapitulated across the Subsets

That there are differences (in inflammatory or recruitment markers) in monocytes (as a whole) between individuals does not negate the concept of monocytes differentiating in the circulation from a classical to nonclassical form and modifying their phenotype in the process. Indeed, a degree of intraindividual subset differences is maintained (as evident in Figure 4). However, the inflammatory state acquired during differentiation is recapitulated across the subsets, as evident by the fact that cytokine production (or surface marker expression) by one subset correlated with that of the next [4]. Thus, though the intermediate subset is broadly considered more inflammatory than the classical, the degree to which it becomes inflammatory depends on the inflammatory state of the classicals. In essence, the baseline inflammatory state (or migratory state [14]) is determined prior to differentiation.

That the increased expression of inflammatory or migratory markers was seen in classical monocytes (including at a single-cell level) suggests that monocytes may be entering the circulation in an inflamed state from the bone marrow. Indeed, this is consistent with current models of haematopoiesis and monocyte differentiation, where precursors adapt to pathological challenges balancing both hierarchical differentiation and cell plasticity [29]. Mitroulis et al., in 2018, were one of the first to hypothesize that the monocyte cellular changes induced during disease also occur in progenitors of the haematopoietic system in the bone marrow [125], with these adaptations likely cascading down to the myeloid lineage cells monocytes/macrophages. Indeed, HSCs mount a transient response to inflammatory stimuli through emergency myelopoiesis (discussed in Section 2.3) which helps fight against the offending trigger. Moreover, the shift towards emergency myelopoiesis releases pro-inflammatory monocytes as characterised by low HLA-DR expression in severe COVID-19 patients in a recent prospective study [126]. The altered baseline inflammatory status of the classical monocytes would then be recapitulated across the ensuing subsets.

### 5.4. Changes in Bone Marrow Cells Cause Persistent Monocyte Inflammation and Account for Different Baseline Monocyte Inflammatory Status

The functional and phenotypic variations across subsets in various infectious and sterile disease conditions may be persistent. This was seen when the inflammatory state of monocytes persevered in people with familial hypercholesterolemia despite three months of statin treatment and the successful lowering of LDL-C [127]. Similarly, a persistent inflammatory phenotype was seen in post-acute myocardial infarction patients after treatment with multiple drugs for 6 months [128]. Foundational evidence of such long-term functional reprogramming of monocytes was gathered from BCG-vaccinated individuals. Monocytes from vaccinated healthy volunteers exhibited enhanced inflammatory phenotype in response to unrelated bacterial and fungal pathogens. Moreover, this monocyte phenotype lasted even three months after vaccination, when all the microorganisms (pertaining to the vaccine) are expected to be cleared from the system [129]. Such long-term functional reprogramming of monocytes (up to months), while initially puzzling as these cells have a short lifespan (1–4 days) in circulation [49], has now been shown to arise through epigenetic modifications. These may not only contribute to long-term altered monocyte function but also enhance the response to secondary stimuli [130]. This was clearly demonstrated in a seminal study on BCG vaccination which led to persistent transcriptomic signatures on human HSPC that induced myeloid development. These transcriptomic changes were associated with specific epigenetic modifications, both of which could be detected in circulating monocytes even 3 months after vaccination [130].

In vivo, the HSC are likely to encounter different infectious stimuli which continually shape the epigenetic landscape of the cells to various degrees. This might add to the complexity of the basal inflammatory status of ensuing monocytes (and monocyte-derived macrophages) released upon each subsequent infection. Such in vivo activation of HSC might also vary across individuals owing to the very nature and duration of exposure to different stimuli during their lifetime, ultimately bringing about the interindividual difference in baseline classical monocyte inflammatory status that is generally observed by us.

### 5.5. Functional Monocyte Changes Are Recapitulated in Macrophages

Circulating monocytes differentiate into mature macrophages, with these monocyte-derived-macrophages having distinct epigenetic imprints of their monocyte or HSC predecessors [131,132]; this makes them functionally distinct from embryonically-derived tissue-resident macrophages [133]. Consequently, re-stimulation of these macrophages within the tissue microenvironment might elicit a distinctive response that is dependent on the individual macrophage ontogeny.

Chronic inflammatory conditions have an added layer to this story. Importantly, chronic inflammation drives a positive feedback loop that continuously activates the haematopoietic progenitors, generating mature myeloid cells with possible increased inflammatory potential [47]. For example, the monocytes induced by several atherogenic risk factors had a persistent activated state in vitro [12] and associated epigenetic rewiring. This points towards the persistent low-grade inflammation found in individuals with chronic atherosclerotic cardiovascular diseases (ASCVDs). The pro-inflammatory monocytes differentiate into pro-inflammatory macrophages that accumulate lipids and form foam cells within the intimal space, contributing towards atherosclerosis progression [134]. Macrophage phenotypes are generally known to be plastic within the tissue microenvironment. However, the subsequent phenotypic switch between inflammatory extremes of M1 (pro-inflammatory) and M2 (healing) macrophages under the conditions of both dysregulated ASCVD risk factors and those favouring resolution of inflammation are unknown and is an area of active investigation. Nevertheless, monocyte-derived-macrophages in chronic conditions bear the lineage-derived epigenetic marks that might leave them leaning towards either a hyperinflammatory or dysregulated healing state when exposed to other stimuli within the tissue microenvironment.

In both acute and chronic inflammatory conditions, the changes (functional, transcriptomic, and epigenetic) induced in HSC are recapitulated to monocytes and their derivative macrophages and this might critically influence the disease progression or resolution.

## 6. Conclusions

Over time, and through improved technology, the heterogeneity of monocytes has been increasingly recognised. Thus, though the three main subsets—classical, intermediate, and nonclassical—have primarily been examined (in health and disease) still further subpopulations within these subsets have more recently been identified. While indeed, monocytes exist on a continuum, changing function as they move from classical to intermediate and nonclassical monocytes, the function of a particular subset, or even sub-population of a subset is not the same between individuals. While this is recognised in a disease with higher counts or altered function of a subset, evidence is emerging that differences between individuals override the differences between the subsets. While this frame of reference has not received a lot of attention in the past, it is consistent with the known plasticity of the myeloid cells, and the more recent understanding that monocytes and even their precursors undergo epigenetic programming in response to their environment. The changes engendered are recapitulated as the cells differentiate. Thus, while particular subsets have been proposed to be specific targets for different diseases, it is more likely to be the altered function of the monocytes overall that needs to be addressed. Identification of individual factors that predispose monocyte precursors and derived cells to specific irreversible functional changes could be targeted for therapy so that the course of the disease could be altered to take on favourable pathways [3].

## Figures and Tables

**Figure 1 ijms-24-08757-f001:**
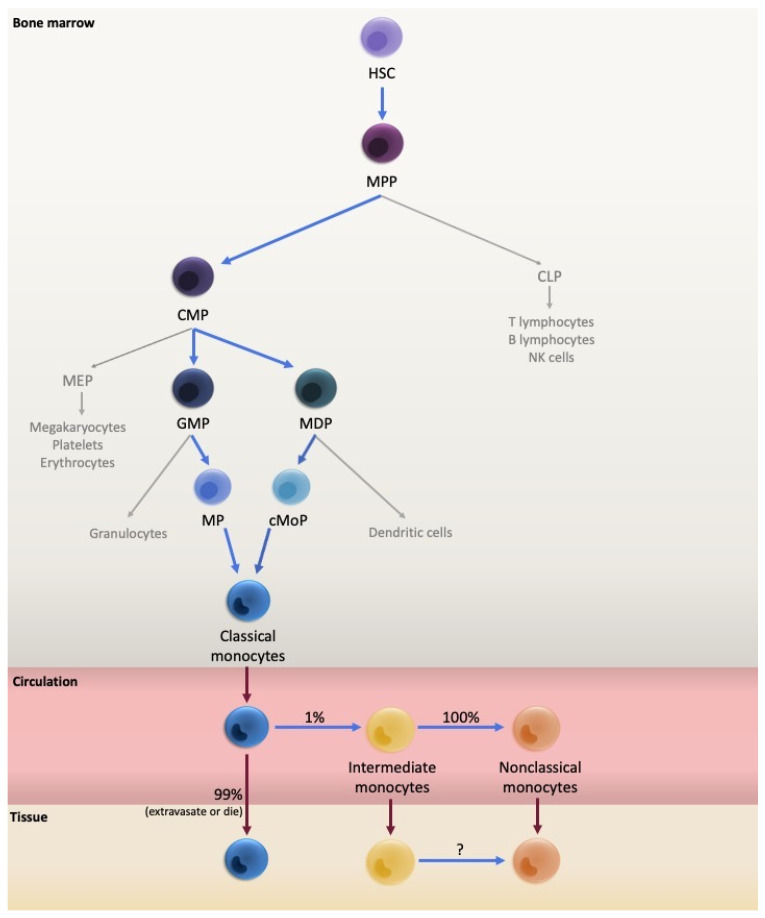
Traditional model of monocyte differentiation and mobilisation. Monocytes arise in the bone marrow from the differentiation of multiple levels of precursors. They are then released into the circulation where they differentiate through the three subsets, which can migrate into tissue. HSC: haematopoietic stem cell; MPP: multipotent progenitor; CMP: common myeloid progenitor; CLP: common lymphoid progenitor; MEP: megakaryocyte-erythrocyte progenitor; GMP: granulocyte-macrophage progenitor; MDP: monocyte-dendritic cell progenitor; MP: monocyte progenitor; and cMoP: common monocyte progenitor.

**Figure 2 ijms-24-08757-f002:**
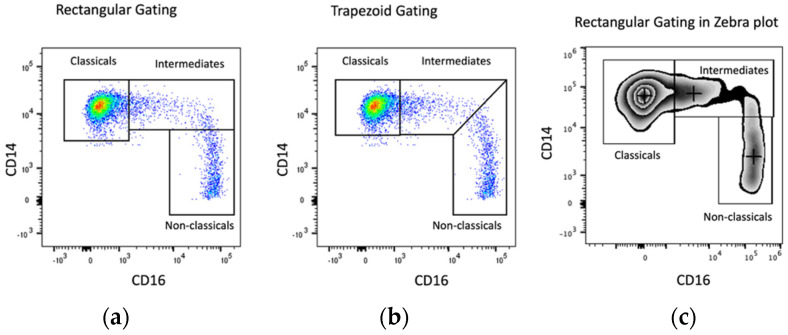
Comparison of gating methods. (**a**) rectangular gating, (**b**) trapezoid gating, (**c**) zebra plot displaying median which allows symmetrical gating of classicals with even distribution of cells around median population.

**Figure 3 ijms-24-08757-f003:**
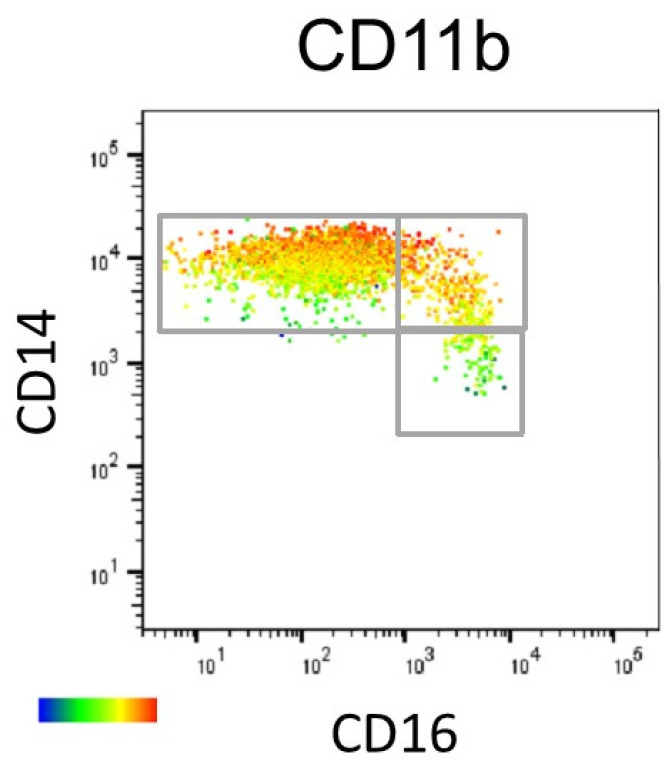
Marker expression within and between subsets. CD11b is expressed highly by some classical monocytes but is low in others, with a similar picture for the intermediate subset. The figure is reprinted with permission from Patel et al. ‘Monocyte subset recruitment marker profile is inversely associated with blood ApoA1 levels’, Frontiers in Immunology 2021 [14]. Copyright 2021 Patel, Williams, Li, Fletcher, and Medbury.

**Figure 4 ijms-24-08757-f004:**
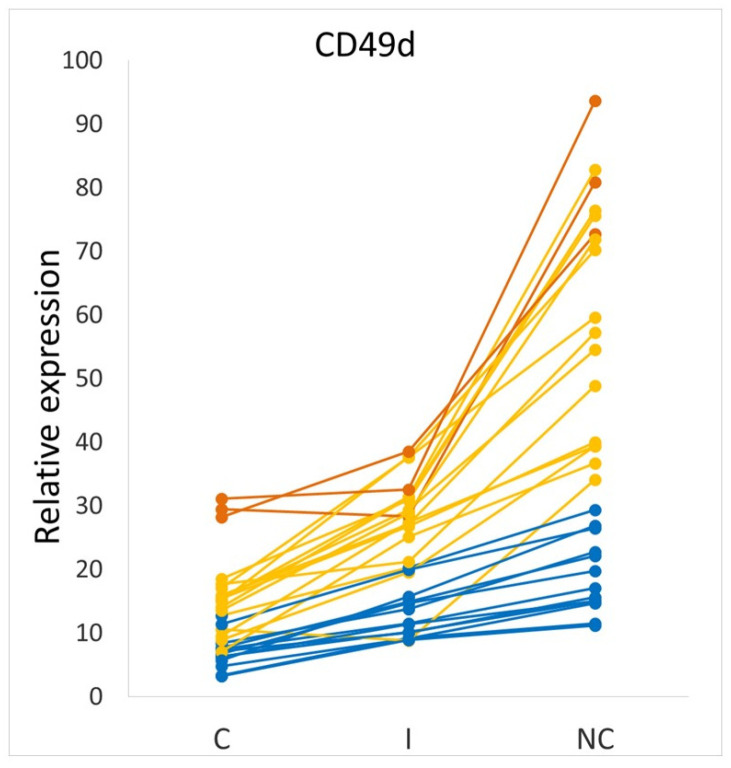
Variation in CD49d between monocyte subsets for different study individuals. Each line on the graph is representative of the relative CD49d marker expression for one individual. Dark orange lines show study subjects with higher expression of CD49d on all three monocyte subsets than the nonclassical subset of the participants indicated by the blue lines. The remainder of the participants’ data is shown in yellow. C: Classical; I: intermediate and NC: nonclassical. The figure is adapted with permission from Patel et al. ‘Monocyte subset recruitment marker profile is inversely associated with blood ApoA1 levels’, Frontiers in Immunology 2021 [14]. Copyright 2021 Patel, Williams, Li, Fletcher, and Medbury.

**Table 3 ijms-24-08757-t003:** Monocyte subset phenotype and function.

	Classical	Intermediate	Nonclassical	References
Inflammation markers	CD64++ CD86+ TNFR1+ TNFR2+ HLADR+	CD64+ CD86++ TNFR1++ TNFR2+ HLADR++	CD64lo CD86+++ TNFR1+ TNFR2++ HLADR+	[4,64,80,88]
Anti-inflammatory Marker	CD163+++ CD36++	CD163++ CD36+	CD163+ CD36-	[9,64]
Chemokine Receptors (Adhesion & Migration)	CCR2++ CCR5+ CX_3_CR1+ CD11b++ CD62L++	CCR2+ CCR5++ CX_3_CR1++ CD11b++ CD62L−	CCR2lo CCR5+ CX_3_CR1+++ CD11b+ CD62L−	[14,64,90,91]
Cytokine & Chemokine Production	Mixed response to LPS (e.g., IL-6, IL-8, CCL2, CCL3)	High in response to LPS (e.g., IL-1β TNFα) Major response to TLR2 agonist	Weak response to LPS Response to viruses, selectively produce TNFα IL-1β, CCL3)	[9]
Overall functions	Phagocytosis Adhesion and migration Antibacterial responses	Phagocytosis Migration role unclear Antibacterial responses Antigen presentation	Weak phagocytosis * Patrolling vasculature, migration Antiviral responses Antigen presentation	

Plus symbols (+, ++, +++) indicate degree of expression. Minus symbol (−) indicates not expressed. * Gene expression indicates complement and FC-mediated phagocytosis. LPS lipopolysaccharide, CCL chemokine (C-C motif) ligand.

**Table 4 ijms-24-08757-t004:** Phenotypic definitions of mouse monocyte subsets.

	Classical	Intermediate	Nonclassical
Identifying marker	Ly6Chi (Gr1+)	Ly6Cint	Ly6Clo (Gr1−)
Chemokine expression	CX_3_CR1+ CCR2+	CX_3_CR1+ CCR2+	CX_3_CR1++ CCR2lo
Human equivalent	CD14++CD16−	CD14++CD16+	CD14+CD16++

References: [10,11].

## Data Availability

No new data were created or analysed in this study.

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
