# Peer review of "Monocyte Differentiation and Heterogeneity: Inter-Subset and Interindividual Differences"

_ijms, 2023, doi:10.3390/ijms24108757_

Round 1

Reviewer 1 Report

- There are references missing in the introduction (line 39, 50 and 51)

- In the introduction the authors state that intermediates and nonclassicals are often described as inflammatory. They made this statement several times throughout the whole manuscript. Although they provide solid evidence for this, it would be good if contradictory results are discussed extensively. Different views regarding this statement will make this review more interesting. 

DOI: 10.1146/annurev-immunol-042617-053119

- Figure 1 supplements paragraph 2.1, however only contains half of the data. More pathways, which are described in the text, may be drawn in the figure. 

- Paragraph 2.2 does not include references on monocyte expansion by proliferation. Hereby, GM-CSF and M-CSF can be mentioned. As well, reverse migration is not mentioned at all. 
Monocyte recruitment during infection and inflammation | Nature Reviews Immunology

- Paragraph 2.2 and chapter 6 should be combined. As ´Monocytes changes start in the bone marrow (title of chapter 6)´ it is more logical to start the review with this chapter. 

- Table 1 is interesting, because of the focus on history. More references to this figure are needed in the text. The last row include data regarding 4 subsets, however only 3 marker strategies are listed. 

- Table 2 is not needed. 

- Explain ´vertical separation´ (line 243)

- Figure 2 is useful. It would be good to also include the pre-gating here. As an example, neutrophils also express CD16. It would be good to mention that those need to be gated out. 

- Which 3 ways (line 308-312)

- Chapter 4 is about ´Monocyte subset functions´, however the authors continue to write about the monocyte spectrum (chapter 3)! For example, table 3 include markers to distinguish subsets. This fits better to chapter 3. Starting from line 405 the focus is more on ´functionality´. Chapter 4 should start here.

- Table 2 should be table 3 (line 351)

- Table 3 is confusing and the references are missing. In my opinion this table can be removed. 

- Paragraph 4 is in general not clear. It contains lots of information, but it is not a coherent story. I would expect a better focus on ´functional assays´ in this chapter. To make it better to understand more explanation is needed. E.g. explain the role of certain genes (line 390 and 391). Why are those of interest? 

- Paragraph 4.1 should be shortened, be more focused and a figure would fit better than a table. Show pathways that are unique or interesting for certain subsets. 

- 450-452 à This paragraph is about phagocytosis and ROS. Suddenly the authors start talking about angiogenic assays and VEGF. This does not fit here.  

- Line 485-487 --> This is contradictory to Ref. 88 

- Paragraph 5.3 should be merged into chapter 6. 

- In chapter 6 the subsets are not mentioned anymore. How does this relate to the rest of the review? 

General: writing needs to be improved and there is some repetition throughout the manuscript. The review will improve by making it shorter and more to the point.

This might be an interesting paper to include:
PD-L1 expression on nonclassical monocytes reveals their origin and immunoregulatory function - PubMed (nih.gov)

The manuscript is readable, but contains many small mistakes. Therefore, extensive editing is required. 

Author Response

  1. There are references missing in the introduction (line 39, 50 and 51)

References have been added to each of these lines (now 40, 49, 53 in tracked version). Note that the references as tracked changes only showed prior to being formatted.  

  1. In the introduction the authors state that intermediates and nonclassicals are often described as inflammatory. They made this statement several times throughout the whole manuscript. Although they provide solid evidence for this, it would be good if contradictory results are discussed extensively. Different views regarding this statement will make this review more interesting. DOI:10.1146/annurev-immunol-042617-053119

We agree that it is more complex than this. We have removed this line from the introduction, instead stating (line 147-tracked) “Researchers have discovered that these subsets show some functional differences, such as their inflammatory responses and migratory potential” We have also removed a line from 4.1 (line 399-tracked) stating “Generally, intermediate and nonclassical subsets were considered to possess enhanced pro-inflammatory functions compared to the classical subset.” and modified wording in 5.3 (line 779-tracked)  to state these subsets are broadly considered more inflammatory.

We thank the reviewer for reminding us of this review paper. We have revisited it and have added a note on anti-inflammatory roles of nonclassicals in atherosclerosis to 4.2 (line 644-tracked), being mindful not to add too much to an already long review.

“As influx of monocytes into atherosclerotic plaques is an important step, classical monocytes are therefore likely to promote atherosclerosis. Nonclassical monocytes with their limited plaque influx are considered atheroprotective, supported by studies showing that mice lacking nonclassical monocytes readily develop atherosclerosis (reviewed [111]).

  1. Figure 1 supplements paragraph 2.1, however only contains half of the data. More pathways, which are described in the text, may be drawn in the figure. 

The figure represents the traditional model, rather than information challenging this model. We have therefore retitled the figure from “Monocyte differentiation and mobilisation.to “Traditional model of monocyte differentiation and mobilisation”. This saves the figure from becoming too complex, particularly considering that the monocyte ontogeny is not a particularly novel section of this review.

  1. Paragraph 2.2 does not include references on monocyte expansion by proliferation. Hereby, GM-CSF and M-CSF can be mentioned. As well, reverse migration is not mentioned at all. 
    Monocyte recruitment during infection and inflammation | Nature Reviews Immunology

We have altered the start of section 2.2 (line 126, tracked) to state “The differentiation of monocytes from HSC, through the progenitor populations described above, predominantly occurs in the adult bone marrow and is promoted by macrophage colony-stimulating factor (M-CSF) [30].”

Further down (line 138, tracked) we have added “Further expansion of the monocyte population in the circulation is supported by the identification and isolation of a subpopulation of human monocytes that proliferate when cultured with M-CSF or granulocyte macrophage colony-stimulating factor (GM-CSF) [34-36].

Towards the end of 2.3 (line 178-tracked) we have added a section on reverse migration “It is also important to note that monocytes not only migrate from the bone marrow to the circulation and then into tissue, but in mice it has been shown that migration also occurs in the reverse direction. Transendothelial migration of monocytes has been observed from the vascular wall directly back into the circulation [39-41] or via drainage to lymph nodes [42]. Furthermore, murine monocytes can even migrate from the circulation back into the bone marrow, to preserve their lifespan during periods of prolonged fasting [43]. “

  1. Paragraph 2.2 and chapter 6 should be combined. As ´Monocytes changes start in the bone marrow (title of chapter 6)´ it is more logical to start the review with this chapter. 

We agree that some of the information in section 6 should have been introduced earlier, so section 6.2 has been moved to become section 2,3. The rest of section 6 has been dissolved into section 5 (see comment 18), so the title of the previous section 6 is no longer present.

  1. Table 1 is interesting, because of the focus on history. More references to this figure are needed in the text. The last row include data regarding 4 subsets, however only 3 marker strategies are listed. 

There was only a single reference to this table in the text so we have acted on this suggestion by adding more references to this table within the text (lines 233, 243 - tracked).

Stating four subsets but only 3 marker strategies was indeed confusing. The paper in question used RNAseq and clustering to identify four monocyte subsets in humans and showed that two of these subsets fell within the currently recognised intermediate subset. To address this, we have added a footnote at “CD14++CD16+” in the table which reads “*Within this subset, single cell RNA-seq identified heterogeneity, namely two distinct clusters which were assigned Mono3 and Mono4. Gene expression profiling showed Mono3 expressed cell cycle and trafficking genes, while Mono4 expressed a cytotoxic gene signature.”

  1. Table 2 is not needed. 

We had intended this table to show that the classification of monocytes into 2, and then 3 subsets, was utilised in numerous studies and shown to have clinical relevance. The table shows that the classifications into two, and then three subsets was adopted into the literature, and provides evidence of consensus.

To make the logic behind inclusion of this table clearer, we have made three changes. First, we have created a clear distinction between Tables 1 and 2 by altering Table 1 header to include “in health”, “Table 1. Categorisation of monocyte subsets in health.” And Table 2 is now titled “Table 2. Utilisation of established monocyte subsets in disease states.” This shows how the two tales differ. Second, we have added a reference to the table early in paragraph 2 section 3.1 and the words “(Table 2). This shows that two subsets was the consensus at this point.” Third, in reference to the naming of the three subsets, we have added “with these names being adopted in clinical research (Table 2).” This ensured in-text reasoning for the table is given.

  1. Explain ´vertical separation´ (line 243)

We have reworded this to “rectangular gating” (line 302- tracked) to match the image in Figure 2a and other sections of the text.

  1. Figure 2 is useful. It would be good to also include the pre-gating here. As an example, neutrophils also express CD16. It would be good to mention that those need to be gated out. 

Considering there are numerous ways to successfully gate monocytes, and that this paper is not focused on the flow cytometry methods, we have elected not to include these pre-gating steps out of concern it focuses too heavily on our approach while others are equally valid. Instead, we have noted in-text the importance of gating out contaminating cells and referred the reader to some reputable papers that do so in different ways.

Line 282- tracked. “Numerous approaches have been employed to get rid of contaminating cells and distinguish the monocytes into three subsets. Monocyte gating steps begin with different ways to eliminate clumps and debris. Next, monocytes are selected either by drawing a tight gate around monocytes in a forward scatter vs. side scatter plot [72,73] or using an additional monocyte specific marker (other than CD14 and CD16) e.g. HLA-DR or CD86. This excludes contaminating cells, particularly neutrophils, T cells, B cells and natural killer cells [74,75].

  1. Which 3 ways (line 308-312)

This sentence read “To further support the idea that monocytes exist not in discrete subsets but as a continuum, a study by Cignarella et. al. (2018) 3 theoretical ways of analysing the monocyte continuum within the traditional CD14/CD16 plot, ultimately paving the way for fine tuning the assessment of CVD risk in large patient cohorts [58].” We agree that to note these but not expand raises questions, so We have expanded to explain

Line 372, tracked. “The first of these three theoretical ways was explored in the study explained above [83], where percentage of cells and MFI of CD14 and CD16 within each gate was reported, to enable exact transitioning monocytes from one subset to another. The second way used multiple, discrete gates along the CD14/CD16 plot, further allowing a more detailed analysis of monocytes as they transition [71,84]. The third way was termed as the clock rule which displays the CD14/CD16 plot as a continuum along a 90° curve, capturing the expression of both markers as a single number. This latter approach is only theoretical and has not been validated yet.”

  1. Chapter 4 is about ´Monocyte subset functions´, however the authors continue to write about the monocyte spectrum (chapter 3)! For example, table 3 include markers to distinguish subsets. This fits better to chapter 3. Starting from line 405 the focus is more on ´functionality´. Chapter 4 should start here.

We appreciate this feedback and have carefully reviewed the section to remove information that relates to spectrum.

Some examples are

Line 410 tracked- removed “Flow cytometric analysis of monocyte subsets indicated that some markers are subset specific which clearly segregates one subset from other subsets, while other markers show a gradual elevation or reduction across the subsets from classicals to non-classicals..”

Line 549 tracked – removed “High dimensional mass spectrometry study has identified subgroups within the subsets which showed variable expression of CD93, CD11a, CD61 and CD9 markers within the classicals and sub-populations that varied in efferocytosis and migration capacity within the non-classical subset.”

For Table 3 we have removed the “identifying markers” row. The other markers, while they differ between subsets, are shown in this context as indicators of function. The broad function being indicated in the row headings e.g. “Inflammation markers”. We have ensured the functions of these markers are described in the text.

  1. Table 2 should be table 3 (line 351)

Thank you for pointing out this error. We removed this line due to it repeating content of the sentence that started the paragraph. The sentence at the start of the paragraph (line 409-tracked) now includes a reference to Table 3  “Monocyte subsets express different surface markers in varying levels reflecting their phenotype and functions (Table 3).”

  1. Table 3 is confusing and the references are missing. In my opinion this table can be removed. 

We feel this table is important but agree it had confusing aspects. Firstly, we have added references in. Secondly, we have simplified the table by removing some information. See also reviewer 2 comment 3 for reasoning behind other changes to this table.

  1. Paragraph 4 is in general not clear. It contains lots of information, but it is not a coherent story. I would expect a better focus on ´functional assays´ in this chapter. To make it better to understand more explanation is needed. E.g. explain the role of certain genes (line 390 and 391). Why are those of interest? 

We have shortened this section by removing information that is not necessary or is covered elsewhere.

Genes and markers mentioned within this section now have functions mentioned. E.g. lines 474, 478 tracked.

15. Paragraph 4.1 should be shortened, be more focused and a figure would fit better than a table. Show pathways that are unique or interesting for certain subsets. 

As stated above, we have shortened this section.

We’ve taken this advice to ensure the table shows unique aspects for each subset but have not created a figure. We felt text better described the differences in functions than would an image in this case.

  1. 450-452 à This paragraph is about phagocytosis and ROS. Suddenly the authors start talking about angiogenic assays and VEGF. This does not fit here.  

We agree this does not fit and have removed this line (line 541-tracked)..

  1. Line 485-487 --> This is contradictory to Ref. 88 

The sentence read On the other hand, CD86 which is associated with inflammatory M1 macrophages [75] is highest on nonclassical monocytes, followed by intermediate and finally classical monocytes, which fits with the inflammatory functions attributed to these subsets [74]. Checking this against reference 88 (Haschka 2022, now ref 104) the authors state MFI of CD86 on classical monocytes was 6100 and in intermediates it was 9500. So this does agree with CD86 being higher on intermediates than classical monocytes. We note that this reference did not present data on nonclassicals.

We suspect the wording was confusing, and have improved this as well as including this reference, so it now reads “In parallel, CD86 which is associated with inflammatory M1 macrophages is highest on nonclassical monocytes, lower on intermediate and lowest on classical monocytes, which fits with the inflammatory functions attributed to intermediate and nonclassical subsets.”

Of note, in the possibility the reviewer meant the contradiction was in the function of CD86, we’ll also address that angle in this response. The cited reference did not assign an inflammatory role to CD86, but this is as the role assigned to cells that were high in CD86 was based on a number of markers being coregulated. The reference we included to support CD86 as an inflammatory marker (Biswas 2012) does clearly link CD86 to M1 macrophages and inflammation. 

  1. Paragraph 5.3 should be merged into chapter 6. 

We understand the reasoning behind this was to improve flow. We have therefore dissolved section 6, with most of it being merged into section 5. Some of section 6.1 now forms the end of section 5.3 (now renamed “Baseline monocyte inflammatory state is determined in the bone marrow and is  recapitulated across the subsets.”, the majority of 6.1 has become 5.4 (renamed “Changes in bone marrow cells cause persistent monocyte inflammation and account for different baseline monocyte inflammatory status.” section 6.3 now forms section 5.5, renamed “Functional monocyte changes are recapitulated in macrophages”.

  1. In chapter 6 the subsets are not mentioned anymore. How does this relate to the rest of the review? 

As section 6 has been dissolved and moved into sections 2 and 5, this should no longer be a concern.

  1. General:writing needs to be improved and there is some repetition throughout the manuscript. The review will improve by making it shorter and more to the point

We have removed some repetition throughout, such as the edits to section 4 (comment 11).

In addition, some other sections we’ve removed to limit repetition include lines 575, 881-884,

  1. This might be an interesting paper to include:
    PD-L1 expression on nonclassical monocytes reveals their origin and immunoregulatory function - PubMed (nih.gov)

This paper focuses on a marker that distinguishes mouse classical and nonclassical monocytes PD-L1, and using this marker allowed the authors to confirm migration roles of nonclassical monocytes already reported in the literature. As such, while we could include it, we do not feel it adds information not already present..

  1. Regarding English, this reviewer noted several small mistakes “Extensive editing of English language required” whereas Reviewer 2 noted “English language fine. No issues detected”.

We have carefully reviewed the article to identify and correct several small errors. These are visible as track changes including  

Examples in lines: 62, 166, 252, 278, 367, 564, 654

Reviewer 2 Report

The manuscript entitled "Monocyte differentiation and heterogeneity in homeostasis and disease" is an another review on this topic and its potential novelty comparing to another similar articles, has not been clearly indicated. The manuscript contains a general knowledge while the relevant details regarding the composition of monocyte subsets and their specific role in diseases are missing. In this context the Authors should introduce several aspects:

1. Dealing with the monocyte subset history the Authors should mention the identification of FcRI+ and FcRI- human monocyte subsets (J. Immunol. 1984, 133, 1293). In this context, further seminal works by Grage-Griebenow, et al. describing CD64-positive and CD64-negative subsets with and w/o CD16 should be mentioned as well. These are important data for the understanding how definition of monocyte heterogeneity evolve, especially that the classification of monocytes for CD64-positive and CD64-negative subsets was valid until the current CD14/CD16.

2. Recently the role of SLAN as a marker of non-classical monocytes has been indicated but this aspect was not sufficiently highlighted in the manuscript. 

3. Phenotype heterogeneity within the monocyte subsets has been only barely addressed and presented mainly with respect to the Authors' previous works. This aspect needs to be described in more detail, referring to the 3 monocyte subsets. In this context the paper by Ozanska et al. could be of reference (Scand.J.Immunol. 2020;92:e12883).

4. The role of monocyte subsets in some diseases has been mentioned but this aspect needs to be better elaborated. The specific disease should be indicated as sub-headings and the role of monocytes/specific subset described. Especially, the critical role of monocytes in atherosclerosis and cancer, with clear cut pro- and anti-cancer activity should be presented.

5. When referring to the mouse studies, the Authors should clearly present (table or graph) the phenotype definition of murine subsets corresponding to those defined in humans.

Minor:

1. title - I would suggest that the Authors substitute "health" for "homeostasis" in the title as a more appropriate. 

2. Abstract - "Studies have revealed monocyte heterogeneity is multidimensional, with their phenotype and function differing between subsets, within each subset, between different groups of people, and even between individuals" - this sentence is misleading. The main subsets possess well defined functions, despite the ethnic group or individuals. Potential differences are related to minor subpopulations.

3. Table 3 - first column is not clearly described - it should refer to the function not to the cellular subset... Moreover, when referring to inflammation the non-classical monocytes are predominantly CD64-negative.  

Round 2

Reviewer 2 Report

The ms. has been corrected and improved. There are some minor issues that need to be fixed.

1. line 21 - "different groups of people" should be replaced by "different ethnic groups";

2. line 571-572 - "...HLA-DR which present a wide range of antigens to CD4 cytotoxic T cells [65, 79] - there is no mention of CD4 cytotoxic T cells in the references cited;

3. the sections "in vitro evidence" and "in vivo evidence" are too long to keep the reader's attention - introducing a few subsections would help.

Author Response

The ms. has been corrected and improved. There are some minor issues that need to be fixed.

  1. line 21 - "different groups of people" should be replaced by "different ethnic groups";

We have discussed as a team and while we agree the initial wording is unclear, we feel this revised wording does not reflect our intention. The differences we intended to mention here were those between health/disease states. Importantly, the differences between ethnic groups were proposed to arise due to exposure to pathogens (from our paper “Though the two populations were different ethnicities, the differences were thought to be attributed to past exposure to parasites rather than genetics.”).

As such, we have amended the wording to reflect the difference between health and disease “However, it is becoming evident that heterogeneity also exists within each subset, between health and disease (current or past) states, and even between individuals.”

  1. line 571-572 - "...HLA-DR which present a wide range of antigens to CD4 cytotoxicT cells [65, 79] - there is no mention of CD4 cytotoxic T cells in the references cited;

We have now added two references to support this function of HLA-DR, but retained the references which showed the high expression of these markers “Intermediates were found to express high levels of class II MHC genes, e.g., CD74, HLA-DR [65,79] which present a wide range of antigens to CD4 cytotoxic T cells [92,93].”

  1. the sections "in vitro evidence" and "in vivo evidence" are too long to keep the reader's attention - introducing a few subsections would help.

We have added subheadings to both of these sections, which explain the content of each subsection. We thank the reviewer, as this guides the reader in what to expect in each subsection and makes it easier to follow.

Other comments

We also corrected a typo of “anti-tumouroral” to “anti-tumoural” on line 554.

Upon re-formatting the reference list, some references that had no longer been cited (in previous version) were automatically removed (refs 130-134 on previous version). This arose as we failed to re-format references after accepting all changes.  

We once again thank the reviewers for their time and valuable feedback.
